# DiffusionSL: Sequence Labeling via Tag Diffusion Process

**Ziyang Huang**♡♠**, Pengfei Cao**♡♠**, Jun Zhao**♡♠**, Kang Liu**♡♠
♡The Laboratory of Cognition and Decision Intelligence for Complex Systems,
Institute of Automation, Chinese Academy of Sciences, Beijing, China
♠School of Artificial Intelligence, University of Chinese Academy of Sciences, Beijing, China
huangziyang2023@ia.ac.cn
{pengfei.cao, jzhao, kliu}@nlpr.ia.ac.cn

## Abstract

**Sequence Labeling (SL)** is long-standing in Natural Language Processing (NLP). Traditionally, discriminative models have been widely used to capture the conditional distribution of sequence tags, rather than generative models. In this paper, we present **DiffusionSL**, a framework that utilizes a conditional discrete diffusion model for generating discrete tag data, resulting in a **Tag Diffusion Process**. We treat the natural language sequence as the conditional signal and the sequence tags as the generation target, iteratively refining the noisy tags to obtain clean ones. To address the discreteness issue, we propose the **B**it-**T**ag **Converter** (**BTConverter**) to model the target in continuous data space. Furthermore, we introduce the **Bit Di**ffusion **T**ransformer (**BitDiT**) to model the process of noise elimination. Leveraging the powerful iterative refinement capability of the diffusion model, **DiffusionSL** achieves superior performance against previous *state-of-the-art* (SOTA) baselines and outperforms gpt-3.5-turbo significantly across multiple benchmark datasets and various tasks [1].

## 1 Introduction

**Sequence Labeling** (**SL**) is a basic paradigm in the natural language processing (NLP) field, which assigns a predefined label to every meaningful unit (e.g., word or character) in a given sequence (He et al., 2020; Lu et al., 2019; Li et al., 2021; Liu et al., 2021). Many NLP tasks belong to this category, such as Named Entity Recognition (NER) (Zhu and Li, 2022; Liu et al., 2022; Shen et al., 2022; Shen et al. 2023; He and Tang 2022; Zhou et al. 2022b), Chinese Word Segmentation (CWS) (Fu et al., 2020; Huang et al., 2021; Maimaiti et al., 2021), Part-Of-Speech (POS) tagging (Zhou et al., 2022a; Nguyen and Verspoor, 2018).

Most current methods tackled SL in a discriminative manner (Akhundov et al., 2018), which usually employed a language model (Hochreiter and Schmidhuber, 1997; Devlin et al., 2019) to encode the sentence and added a token-level classifier to capture the conditional tag distribution (Zhang and Yang, 2018; Yan et al., 2019; Li et al., 2020; Cui et al., 2021; Wu et al., 2022).

Recently, generative pre-trained language models have demonstrated their versatility in various NLP tasks (Raffel et al., 2020). In consequence, researchers attempt to formulate SL as a sequence generation problem (Athiwaratkun et al., 2020; Paolini et al., 2021; Yan et al., 2021; Lu et al., 2022). This paradigm offers the advantage of flexible generation formats (Raman et al., 2022), allowing models to be trained to produce output in any desired format using teacher forcing. In these generative approaches, the input sentence is encoded first, and the label sequence is then decoded in an autoregressive manner, as depicted in Figure 1(a). Nevertheless, the intrinsic token-by-token generation style of the autoregressive model results in inefficient inference, and the difference in conditional signal during training and inference leads to the exposure bias problem (Arora et al., 2022).

To this end, this paper resorts to the Diffusion Probabilistic Model (DPM) (Sohl-Dickstein et al., 2015; Ho et al., 2020; Nichol and Dhariwal, 2021; Song and Ermon, 2019; Song et al., 2021b), renowned as one of the most powerful generative models in the field of Artificial Intelligence Generated Content (AIGC) (Cao et al., 2023; Wu et al., 2023a), especially in the realm of image synthesis (Liu et al.; Rombach et al., 2022; Hong et al., 2022). DPM gradually introduces noises into clean data through a predefined forward process and subsequently employs a denoising network in a reverse process to recover the original data from pure Gaussian noise. The generation process operates in parallel. Compared with an autoregressive decoder,

---

[1] Code is available at `https://www.github.com/hzy312/DiffusionSL`

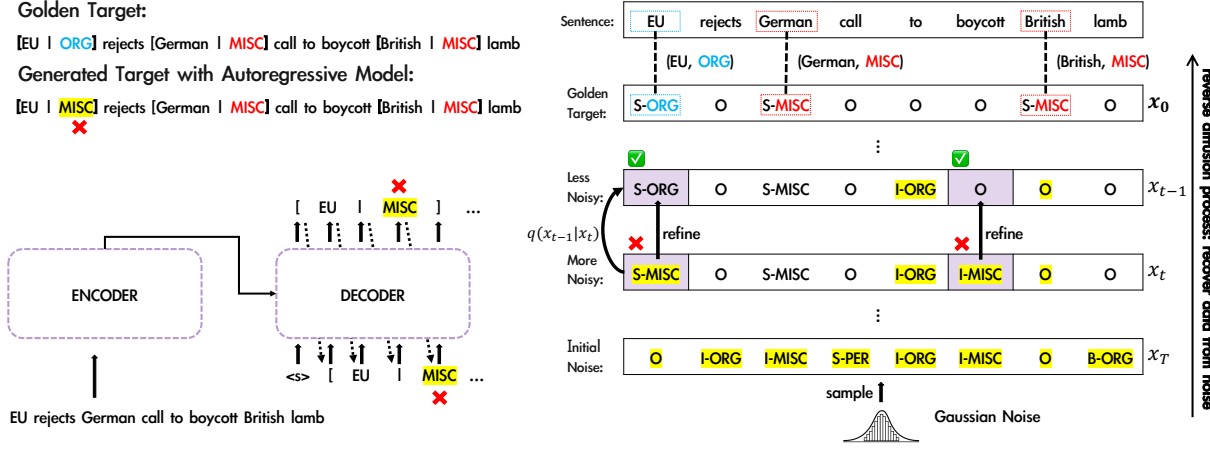

**(a) Autoregressive sequence target generation process**

**(b) Non-Autoregressive sequence target diffusion process**

Figure 1: (a) shows one traditional autoregressive generative method (Paolini et al., 2021) for one typical SL task, Named Entity Recognition (NER). This line of methods generates the target at once and can not refine them. (b) demonstrates our non-autoregressive stepwise generative framework DiffusionSL for NER. In inference stage, the reverse diffusion process of DiffusionSL utilizes a well-trained denoising network to remove the noise step-by-step and recover the clean tags finally, allowing contents (task-specific tags) refinement in this process. Symbols marked in yellow background are noisy, whereas symbols without yellow background represent the desired generated ones.

it could naturally solve the exposure bias problem. Furthermore, the diffusion model can generate the targets progressively to refine the contents step-by-step, thereby facilitating the correction of some mistakes during the generation process. In comparison, the conventional autoregressive generative model generates the targets at once. The aforementioned advantages contribute to the success of the diffusion model in generating data across various modalities (Zhang et al., 2023a; Trippe et al., 2023; Li et al., 2022c; Li et al., 2022b). Intuitively, we expect that DPM could also present superior performance on SL tasks.

Therefore, this paper proposes a novel framework called DiffusionSL. The input sentence serves as the conditional signal, while the corresponding task-related tag sequence is the generation target. We call the entire diffusion process as the Tag Diffusion Process. During the training phase, a fixed forward process is utilized to sequentially perturb the tags and a reverse diffusion process is learned to eliminate noise based on the noisy tags and the encoded sentence. In the inference phase, we sample noisy tags from a standard Gaussian distribution and employ the well-trained denoising model to refine these tags into clean ones in an incremental manner (shown in Figure 1(b)).

However, directly applying a DPM on SL task is challenging. Classical DPM models the image data and Gaussian noise in a continuous space. Instead, tags in SL are discrete. A few research ef-

forts have addressed this issue (e.g., text symbols (Gong et al., 2023), segmentation maps (Chen et al., 2023b, 2022)). Following them, this paper introduces a lightweight and flexible module named Bit-Tag Converter (BTConverter) to represent the tags as binary bits, which can be viewed as real numbers in continuous space. Additionally, we propose the Bit Diffusion Transformer (BitDiT) to model the noise-removal reverse process, which enjoys faster convergence and stronger performance.

Currently, Shen et al. (2023) also introduces a diffusion model into NER, an important SL task. However, their approach only models the diffusion process on entity span boundaries, restricting itself to NER and impeding the potential for extension to other SL tasks.

In summary, our contributions are as follows:

- This study proposes a novel framework DiffusionSL that addresses the Sequence Labeling (SL) tasks using a non-autoregressive stepwise generative approach. To the best of our knowledge, we are the first to apply the diffusion model to SL rather than only NER.

- We introduce two key modules, BTConverter and BitDiT, which convert the discrete data into continuous bits and model the iterative denoising process. The first module effectively handles the discreteness issue while the second one accelerates the convergence.

- We utilize the DiffusionSL for NER, CWS, and POS tagging. The thorough experiments on several datasets of these typical tasks indicate that DiffusionSL can achieve better than previous SOTAs, highlighting the superiority of the proposed Tag Diffusion Process.

## 2 Preliminary

The diffusion model is characterized as a latent generative model. It encompasses a forward process that involves the gradual introduction of noise to clean data $\mathbf{x}_0$. Additionally, it includes a reverse process that incrementally eliminates the noise, thereby generating data following the original distribution. The forward process $q(\mathbf{x}_t|\mathbf{x}_{t-1})$ can be mathematically expressed as:

$$q(\mathbf{x}_{1:T}|\mathbf{x}_0) = \prod_{t=1}^{T} q(\mathbf{x}_t|\mathbf{x}_{t-1}) \qquad (1)$$

$$q(\mathbf{x}_t|\mathbf{x}_{t-1}) \sim \mathcal{N}(\sqrt{1-\beta_t}\mathbf{x}_{t-1}, \beta_t\mathbf{I}) \qquad (2)$$

Here, $\mathcal{N}$ represents the Gaussian distribution and $\beta_t$ represents the noise introduced at timestep $t$. The magnitude of the noise is governed by a noise schedule $\beta = \{\beta_t\}_{t=1}^{T}$. Generally, the noise intensity usually follows either a linear (Ho et al., 2020) or cosine (Nichol and Dhariwal, 2021) function. In this study, we adopt the linear noise schedule. By employing the notations $\alpha_t = 1 - \beta_t$ and $\bar{\alpha}_t = \prod_{s=1}^{t} \alpha_s$, we can deduce an equation to directly incorporate noise into $\mathbf{x}_0$, yielding the noisy data at any given timestep:

$$q(\mathbf{x}_t|\mathbf{x}_0) \sim \mathcal{N}(\sqrt{\bar{\alpha}_t}\mathbf{x}_0, (1-\bar{\alpha}_t)\mathbf{I}) \qquad (3)$$

Additionally, the reverse process $q(\mathbf{x}_{t-1}|\mathbf{x}_t)$ allows us to reconstruct samples from pure Gaussian noise, which is not tractable directly. We can use a neural network $p_\theta$ to estimate it:

$$p_\theta(\mathbf{x}_{0:T}) = p(\mathbf{x}_T) \prod_{t=1}^{T} p_\theta(\mathbf{x}_{t-1}|\mathbf{x}_t) \qquad (4)$$

$$p_\theta(\mathbf{x}_{t-1}|\mathbf{x}_t) \sim \mathcal{N}(\boldsymbol{\mu}_\theta(\mathbf{x}_t, t), \boldsymbol{\Sigma}_\theta(\mathbf{x}_t, t)) \qquad (5)$$

The conditional distribution of $\mathbf{x}_{t-1}$ given $\mathbf{x}_t$ in reverse process could be obtained analytically as follows when $\mathbf{x}_0$ is available:

$$q(\mathbf{x}_{t-1}|\mathbf{x}_t, \mathbf{x}_0) \sim \mathcal{N}(\tilde{\boldsymbol{\mu}}_t(\mathbf{x}_t, \mathbf{x}_0), \tilde{\beta}_t\mathbf{I}) \qquad (6)$$

$$\tilde{\boldsymbol{\mu}}_t = \frac{\sqrt{\bar{\alpha}_{t-1}}\beta_t}{1-\bar{\alpha}_t}\mathbf{x}_0 + \frac{\sqrt{\alpha_t}(1-\bar{\alpha}_{t-1})}{1-\bar{\alpha}_t}\mathbf{x}_t \qquad (7)$$

$$\tilde{\beta}_t = \frac{1-\bar{\alpha}_{t-1}}{1-\bar{\alpha}_t}\beta_t \qquad (8)$$

Therefore, we train the $\boldsymbol{\mu}_\theta$ to predict the $\tilde{\boldsymbol{\mu}}_t$. Due to $\mathbf{x}_t = \sqrt{\bar{\alpha}_t}\mathbf{x}_0 + \sqrt{1-\bar{\alpha}_t}\epsilon$, we could get $\boldsymbol{\mu}_\theta$ by predicting it directly or predicting $\epsilon$, $\mathbf{x}_0$ indirectly. The variance $\boldsymbol{\Sigma}_\theta$ could be learned or fixed. We keep it fixed in this work. The training objective is the variational lower bound of the negative log-likelihood $-\log p_\theta(\mathbf{x}_0)$, which can be simplified to predict the clean data $\mathbf{x}_0$ or injected noise $\epsilon$.

To expedite inference, it is possible to skip certain timesteps in the reverse process. By taking in the more noisy samples at timestep $t_i$, we can generate the less noisy samples at timestep $t_{i-1}$:

$$p_\theta(\mathbf{x}_{t_{i-1}}|\mathbf{x}_{t_i}) \sim \mathcal{N}(\tilde{\boldsymbol{\mu}}, \tilde{\boldsymbol{\Sigma}}) \qquad (9)$$

$$\tilde{\boldsymbol{\mu}} = \sqrt{\bar{\alpha}_{t_{i-1}}}\tilde{\mathbf{x}}_0 + $$

$$\sqrt{1-\bar{\alpha}_{t_{i-1}} - \sigma_{t_i}^2}\frac{\mathbf{x}_{t_i} - \sqrt{\bar{\alpha}_{t_i}}\tilde{\mathbf{x}}_0}{\sqrt{1-\bar{\alpha}_{t_i}}} \qquad (10)$$

$$\tilde{\boldsymbol{\Sigma}} = \eta\tilde{\beta}_{t_i}\mathbf{I} \qquad (11)$$

where $\tilde{\mathbf{x}}_0$ represents the predicted original data by a learned network. $\eta$ is usually set to 0, thus resulting in a deterministic generation process which is called DDIM sampling (Song et al., 2021a).

## 3 Methodology

The details of DiffusionSL are illustrated in Figure 2. We will provide a comprehensive explanation of it in this section. Firstly, we formally describe the overall procedure of the Tag Diffusion Process employed by DiffusionSL. Secondly, we present a detailed description of two components we proposed to help modeling the Tag Diffusion Process: BTConverter and BitDiT. Lastly, we summarize the training and inference algorithms of DiffusionSL.

### 3.1 Tag Diffusion Process

Tag Diffusion Process shown in Figure 2(a) describes a conditional generation process where the conditional signal is input sentence $\mathbf{W} = (w_1, \ldots, w_n)$ and the target is the task-specific tags $\mathbf{L} = (\ell_1, \ldots, \ell_n)$. The set of all possible tags is denoted as $\mathcal{S}$, e.g., {B, M, E, S} for Chinese Word Segmentation. Initially, we construct indices for tags in $\mathcal{S}$ and transform them into bits in continuous space using BTConverter. The resulting transformed clean tag data is referred to as $\mathbf{x}_0$. Noise is incrementally injected into $\mathbf{x}_0$ during the forward diffusion process, while a denoising network is learned to gradually eliminate the noise during the reverse diffusion process, ultimately generating the desired tag data.

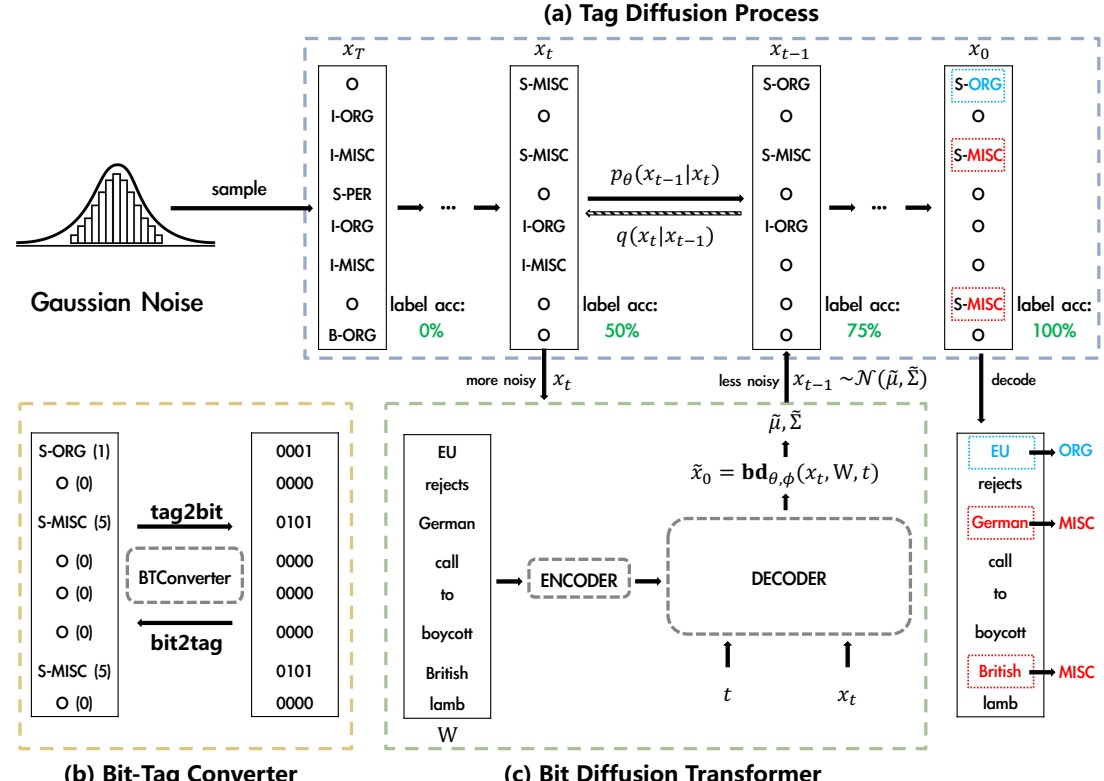

Figure 2: The overall picture of our framework DiffusionSL. (a) depicts the Tag Diffusion Process that eliminates the noise in the tag data in a progressive manner, which transforms the noisy tags sampled from a Gaussian distribution into clean tags. In the actual scenario, the tags in this diffusion process are in bit representation, we convert them into discrete tag representation for clarity in illustrating the denoising process. (b) demonstrates the Bit-Tag Converter (BTConverter) which enables flexible conversion between the discrete tags and the continuous bits. (c) presents the Bit Diffusion Transformer (BitDiT), the denoising neural network.

Specifically, we follow the Equation (3) to perturb the data for the forward diffusion process. More specifically, noisy tags $\{\mathbf{x}_t\}_{t=1}^T$ at each noise level could be obtained by the following equation:

$$q(\mathbf{x}_t|\mathbf{x}_0) = \sqrt{\bar{\alpha}_t}\mathbf{x}_0 + (1 - \bar{\alpha}_t)\epsilon \qquad (12)$$

During the reverse diffusion process, we sample an incomplete timestep sequence of length $\lambda$ to generate the target tags. We train a denoising network, denoted as $\mathbf{bd}_{\theta,\phi}$, which takes in the noisier tags $\mathbf{x}_{t_i}$ at the timestep $t_i$ along with the input sentence $\mathbf{W}$ and then outputs the predicted $\tilde{\mathbf{x}}_0$. Here, $\mathbf{bd}$ is the abbreviation for BitDiT. Further details on this network will be provided in Section 3.3. Subsequently, we can obtain the less noisy tags $\mathbf{x}_{t_{i-1}}$ following the Equation (9), (10) and (11):

$$\mathbf{x}_{t_{i-1}} = \sqrt{\bar{\alpha}_{t_{i-1}}}\mathbf{bd}_{\theta,\phi}(\mathbf{x}_{t_i}, \mathbf{W}, t_i) + \sqrt{1 - \bar{\alpha}_{t_{i-1}} - \sigma_{t_i}^2}\frac{\mathbf{x}_{t_i} - \sqrt{\bar{\alpha}_{t_i}}\mathbf{bd}_{\theta,\phi}(\mathbf{x}_{t_i}, \mathbf{W}, t_i)}{\sqrt{1 - \bar{\alpha}_{t_i}}} \qquad (13)$$

By iteratively performing $\lambda$ denoising steps, the desired target tags are acquired.

### 3.2 Bit-Tag Converter

The diffusion model is unable to generate the discrete data directly. Thus, we propose Bit-Tag Converter (BTConverter) to transform the discrete sequence tags into the bits during training (tag2bit) and convert the bits back to the discrete tags during decoding (bit2tag). The functionality of the BTConverter is illustrated in Figure 2(b). We construct an index for every tag in $\mathcal{S}$. If the number of elements in the tag set is $|\mathcal{S}|$, we use $\lceil\log_2|\mathcal{S}|\rceil$ bits to represent tags. We assign $0 \sim \mathcal{S} - 1$ to every tag in $\mathcal{S}$ sequentially, and the bit representation of the $i$-th tag is its corresponding binary bits (i.e., $(101)_2$ represents the 5-th tag), which could be denoted as tag2bit($i$). After converting the tags into bits, every bit is taken as an independent real number. Then, they are shifted and scaled to the range of $[-b, b]$, where $b$ represents a hyper-parameter named signal-noise-ratio (SNR) (Xu et al., 2023; Ji

**Algorithm 1** Training for DiffusionSL

```
# input_ids, attention_mask: [bsz, len]
# gold_seq_tags: [bsz, len]

# encode conditional signal with a PLM
hid = encoder(input_ids, attention_mask)

# convert tags into continuous bits
# [bsz, seq_len, num_bits]
tag_bits = tag2bit(gold_seq_tags)
tag_bits = (tag_bits * 2 - 1) * SNR

t = Uniform(1, T)
epsilon = Normal(0, 1)
crpt_bits = sqrt(alpha_bar(t)) * tag_bits +
    sqrt((1 - alpha_bar(t))) * epsilon

pred_bits = bd(corrupt_bits, t, hid)
loss = mse_loss(pred_bits, tag_bits)

return loss
```

**Algorithm 2** Inference for DiffusionSL

```
# input_ids, attention_mask: [bsz, len]
# encode conditional signal with PLM
hid = encoder(input_ids, attention_mask)

# begin with an isotropic Gaussian
bits = Normal(0, 1)

# denoise the data iteratively
for t in range(T):
    bits_0_pred = bd(bits, t, hid)
    bits = ddim_sample(bits_0_pred)

tags = bit2tag(bits)

return tags
```

et al., 2023). After the inference with the reverse diffusion process, we use $\texttt{bit2tag}(bits)$ to convert bits back into discrete tags, which are then decoded based on the schema of the downstream task. Further details of the $\texttt{tag2bit}$ and $\texttt{bit2tag}$ functions are provided in Appendix A.

### 3.3 Denoising Network: Bit Diffusion Transformer

To eliminate noise during the reverse diffusion process, we employ a denoising network based on the Transformer architecture (Vaswani et al., 2017). This network, named Bit Diffusion Transformer (BitDiT), takes the bit representation of the noisy tags $\mathbf{x}_t$, the input sentence $\mathbf{W}$, and the timestep $t$ as inputs and generates the predicted clean tags $\tilde{\mathbf{x}}_0$. The architecture of BitDiT is illustrated in Figure 2(c). It consists of a BERT-style (Devlin et al., 2019) encoder that represents the sentence in hidden space and a decoder that generates the targets in parallel. The encoder and decoder are parameterized by $\theta$ and $\phi$ respectively. The overall functionality of the denoising network could be represented as follows:

$$\tilde{\mathbf{x}}_0 = \mathbf{bd}_{\theta,\phi}(\mathbf{x}_t, \mathbf{W}, t) \qquad (14)$$

In more detail, we first obtain the encoding vectors of the input sentence and cache them for all inference timesteps:

$$\mathbf{H} = \text{Encoder}_\theta(\mathbf{W}) \qquad (15)$$

Next, we use the decoder to generate the clean tags. The decoder comprises four components:

**Condition Embedder**: It encodes the timestep $t$ using a learned lookup table and fuses it with the cached $\mathbf{H}$ to get the conditional signal $\mathbf{c}_t$, allowing the decoder to be aware of input and time:

$$\mathbf{c}_t = \mathbf{Linear}(\text{Concate}(\mathbf{H}, \text{Embeds}(t))) \quad (16)$$

**Pre Layer**: This layer consists of a multi-layer perceptron (MLP) that projects the bit representations to a higher dimension for modeling more fine-grained interactions between tags.

**Generator**: This component consists of $N$ our designed Transformer blocks. It generates clean tags based on the conditional signal $\mathbf{c}_t$.

**Post Layer**: This layer performs layer normalization to normalize the representations and then applies another MLP to project them back to the original dimension.

The generation process can be summarized as:

$$\tilde{\mathbf{x}}_0 = \mathbf{Post}(\mathbf{Generator}(\mathbf{Pre}(\mathbf{x}_t), \mathbf{c}_t)) \qquad (17)$$

where $\mathbf{Pre}$ and $\mathbf{Post}$ represents the Pre Layer and the Post Layer. Instead of using cross-attention to incorporate the conditional signal in the Transformer blocks of Generator, we resort to adaptive layer normalization (Perez et al., 2018; Brock et al.; Li et al., 2022a; Peebles and Xie, 2022), which predicts the element-wise affine parameters based on the condition $\mathbf{c}_t$ to substitute the naive layer normalization (Ba et al., 2016) used in the original Transformer (Vaswani et al., 2017). The layer normalization used in the Post Layer has also been modified to this accordingly. We also learn an additional scale factor for this Transformer before each skip connection (He et al., 2016) to control the amount of newly learned information added. We refer the architecture diagram and calculation details of this Transformer block in Appendix B.

| Methods | MSRA | | | Resume | | | Conll03 | | |
|---|---|---|---|---|---|---|---|---|---|
| | P | R | F | P | R | F | P | R | F |
| *discriminative* | | | | | | | | | |
| BiLSTM-tagger | 90.74 | 86.96 | 88.81 | 93.66 | 93.31 | 93.48 | - | - | - |
| Zhang and Yang (2018) | 93.57 | 92.79 | 93.18 | 94.81 | 94.11 | 94.46 | - | - | - |
| Yan et al. (2019) | - | - | 92.74 | - | - | 95.00 | - | - | 91.45 |
| Li et al. (2020) | - | - | 94.12 | - | - | 95.45 | - | - | - |
| Wu et al. (2022) | 94.92 | 94.19 | 94.55 | 95.63 | 95.52 | 95.58 | - | - | - |
| *generative* | | | | | | | | | |
| Athiwaratkun et al. (2020)* | - | - | - | - | - | - | - | - | 92.00 |
| Paolini et al. (2021)* | - | - | - | - | - | - | - | - | 91.48 |
| Lu et al. (2022)* | - | - | - | - | - | - | - | - | 92.17 |
| DiffusionNER$^\triangle$ | 95.71 | 94.11 | 94.91 | - | - | - | 92.99 | 92.56 | 92.78 |
| DiffusionNER$^\dagger_{\text{reproduction}}$ | (95.64) | (93.97) | (94.80) | (96.64) | (91.84) | (94.18) | (92.35) | (92.76) | (92.55) |
| gpt-3.5-turbo$^\ddagger$ (one-shot) | 48.03 | 57.47 | 52.33 | 64.74 | 28.16 | 39.25 | 34.67 | 61.75 | 44.41 |
| **DiffusionSL** (Ours) | 95.69 | 95.28 | **95.49** | 96.58 | 96.22 | **96.40** | 93.15 | 92.26 | **92.70** |

Table 1: Results of NER. * demonstrates the results of the autoregressive generative methods, which exhibit inferior performance compared to the non-autoragressive ones. The results of DiffusionNER (Shen et al., 2023) on Resume dataset were not reported in the original paper, so we reproduce and test on it, along with the other two datasets. $\triangle$ represents the results reported in the original DiffusionNER paper while the $\dagger$ refers to our reproduced results. $\ddagger$ is the results obtained by prompting gpt-3.5-turbo with one demonstration example.

### 3.4 Training and Inference

For training, we use the Mean Squared Error Loss (MSE Loss) to regress the predicted bits:

$$\mathcal{L}_{\text{MSE}} = \mathbb{E}_{\mathbf{x}_0, \mathbf{W}, t, \epsilon} ||\mathbf{x}_0 - \mathbf{bd}_{\theta, \phi}(\mathbf{x}_t, \mathbf{W}, t)||_2^2 \quad (18)$$

For inference, we sample a sequence of tags in bit representation with the same length as the input sentence from a standard Gaussian distribution. Then, we conduct $\lambda$ denoising steps to get the desired clean bits. The discrete tags are acquired by the `bit2tag` function of the BTConverter. The pseudo-code of training and inference is provided in Algorithm 1 and Algorithm 2, respectively.

## 4 Experiments

### 4.1 Experimental Settings

We conduct experiments to evaluate the proposed DiffusionSL framework on seven different datasets. We choose MSRA (Levow, 2006), Resume (Zhang and Yang, 2018), and Conll03 (Tjong Kim Sang and De Meulder, 2003) for NER, MSRA, PKU (Emerson, 2005), and CTB6 (XUE et al., 2005) for CWS, and CTB5 (XUE et al., 2005) for POS tagging. The detailed statistics of these datasets are provided in Appendix C. We provide the detailed hyper-parameter settings in Appendix D.

We compare the previous models for NER, CWS, and POS tagging to test the effectiveness of the DiffusionSL. Compared methods can be divided into discriminative and generative ones. The former

capture the conditional tag distribution and the latter generates the desired outputs at the decoder end. For NER baselines, the works of Huang et al. (2015a), Zhang and Yang (2018), Yan et al. (2019), Li et al. (2020) and Wu et al. (2022) are discriminative while the works of Athiwaratkun et al. (2020), Paolini et al. (2021), Lu et al. (2022) and Shen et al. (2023) are generative. CWS (Yang et al., 2017; Ma et al., 2018; Yang et al., 2019; Tian et al., 2020) and POS (Diao et al., 2020; Meng et al., 2019) baselines are all discriminative ones. We also compare one of the most powerful Large Language Models (LLMs), gpt-3.5-turbo (OpenAI, 2023). More details of baselines are deferred to the Appendix E.

### 4.2 Overall Results

DiffusionSL achieves better performance on various datasets across different tasks. Promising results validate the effectiveness of DiffusionSL over the previous baselines, including both discriminative and generative methods. NER results in Table 1 show that DiffusionSL achieves 95.49, 96.40, and 92.70 F1 scores on MSRA, Resume, and Conll03 datasets. Meanwhile, for CWS, DiffusionSL achieves 98.33, 96.54, 97.46 F1 scores on MSRA, PKU, and CTB6 respectively according to Table 2. For POS tagging, our method achieves 97.18 F1 score on CTB5, with an improvement of 0.36, which is demonstrated in Table 2. The strong performance proves this non-autoregressive stepwise generative approach can generate task-specific tags effectively and thus tackle the corresponding

| CWS | MSRA | PKU | CTB6 |
|---|---|---|---|
| Ma et al. (2018) | 98.10 | 96.10 | 96.70 |
| Yang et al. (2019) | 97.80 | 95.90 | 96.30 |
| Tian et al. (2020) | 98.16 | 96.47 | 97.13 |
| Cui et al. (2021)[†] | 98.31 | 96.51 | 97.39 |
| **DiffusionSL**(Ours) | **98.33** | **96.54** | **97.46** |

| POS tagging | CTB5 |
|---|---|
| Diao et al. (2020) | 96.64 |
| Meng et al. (2019) | 96.61 |
| Cui et al. (2021)[†] | 96.82 |
| **DiffusionSL** (Ours) | **97.18** |

Table 2: Results of CWS and POS tagging. † denotes the results reproduced by us.

| Shot | 1 | 5 | 10 | 15 |
|---|---|---|---|---|
| F1 | 41.78 | 44.49 | **46.73** | 39.25 |

Table 3: Gpt-3.5-turbo experiments results on Resume dataset. Shot is the number of demonstration example used in prompting. Due to the budget of OpenAI api, we only test one dataset.

problems effciently.

Compared to DiffusionNER, which is our concurrent work that applies the diffusion model on NER, we surpass it on all the datasets, illustrating the advantages of Tag Diffusion Process over the entity span boundary diffusion process. In our view, the generation format (e.g., span boundary) is unnatural, and it separates the span localization and the entity label classification to some extent. Hence the diffusion model can not model the entity label information well. Besides, this generation format necessitates extra heuristic post-processing operations. As a result, it limits this framework to the NER task and results in sub-optimal performance. Our DiffusionSL can not only handle NER but also all other SL tasks, which enjoys stronger performance and more flexibility at the same time.

We also compare the gpt-3.5-turbo (OpenAI, 2023), which is one the most powerful LLMs among OpenAI gpt-series, on NER datasets. We prompt the LLM with one demonstration example (one-shot) for three NER datasets. The experiment of adding more demonstrations is also conducted on Resume dataset. Our experimental results in Table 3 test that the performance will not increase infinitely with the increase of demonstrations. We find that LLM still could not catch up with the task-specific supervised small models. Thus exploring the specific algorithms for sequence labeling is still meaningful. The corresponding prompt details are

| | **Bit** | Embedding$_{\#bits}$ | Embedding$_{768}$ |
|---|---|---|---|
| NER-Resume | 96.40 | 50.07 | 96.28 |
| POS-CTB5 | 97.18 | 96.72 | 96.86 |

Table 4: Ablation study of tag representation on Resume NER dataset and CTB5 POS tagging dataset. #bits is the bit representation dimension.

| | 1 | **10** | 50 | 100 |
|---|---|---|---|---|
| NER-Resume | 96.37 | 96.40 | 96.40 | 96.40 |
| POS-CTB5 | 97.18 | 97.18 | 97.19 | 97.19 |

Table 5: Ablation study of denoising sampling steps on Resume NER dataset and CTB5 POS tagging dataset.

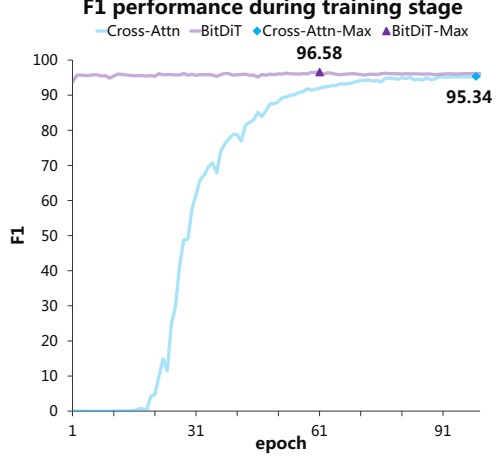

Figure 3: The convergence process of Cross-Attn and BitDiT. Here, *-Max denotes the best performance of method *. The best performance of BitDiT differs from the result reported in Table 1, as the latter presents the test F1 score corresponding to the highest dev F1.

shown in Appendix F.

## 5 Ablation Study and Analysis

### 5.1 Effects of BTConverter

We conduct experiments to compare with the most intuitive and straightforward alternative, embedding method, which creates a learnable embedding table for each discrete symbol in $\mathcal{S}$. Results are shown in Table 4. During training, a prototype embedding is learned for each tag. In inference phase, the embeddings generated from the diffusion model are decoded into discrete tags based on their distance to different prototypes. For fair comparison, we firstly set the dimension of the embedding table the same as the bit representation dimension. Resume has 13 tags and CTB5 has 103 tags in $\mathcal{S}$. Their corresponding bit dimension is 4 and 7. The embedding representation exhibits inferior performance compared to the bit

representation. Even if we scale the dimension of embedding representation to 768, it still can not outperform bit representation and requires additional $|\mathcal{S}| \times h_{tag}$ trainable parameters. Additionally, the decoding method requires calculating the distance from each tag embedding to each prototype, which is more complex than the `bit2tag` function in BT-Converter that only operates once for every tag during decoding. These results demonstrate the lightweight and flexible nature of BTConverter.

## 5.2 Effects of BitDiT

BitDiT incorporates conditional signals through adaptive layer normalization, whereas the standard Transformer decoder block and the DiffusionNER utilize an additional cross-attention layer to perceptualize the conditional information. We refer to this method as Cross-Attn (Vaswani et al., 2017; Shen et al., 2023). Figure 3 shows the convergence process of BitDiT and Cross-Attn on the Reusume dataset. From the result, we discover that Cross-Attn exhibits slower convergence speed and inferior final performance compared to BitDiT. This finding underscores the significance of our proposed BitDiT architecture.

## 5.3 Effects of Denoising Steps

We validate the impact of denoising steps on performance, while the corresponding results are shown in Table 5. We find that the performance improves with the increase of the steps until reaching the bottleneck. Owing to the iterative refinement nature of the diffusion model, we can increase the sampling steps to boost the performance (Chen et al. 2022, Shen et al. 2023). But the performance will not improve infinitely with the increase of the steps. So we choose 10 as the defaulting steps for balance between the performance and the speed.

## 6 Case Study

We visually show the denoising process details by one Chinese NER case. In Figure 4, we input the sentence 中国中央致中国致公党十一大的贺词 (The CPC Central Committee's congratulatory message to the 11th National Congress of the Chinese Zhigong Party) as condition signal, then we generate the corresponding tags in ten decoding steps by iterative refinement. We also show the label accuracy of tags in the bottom for every step, which increases with the number of reverse steps. Finally, we get the exactly correct NER tags of the input

sentence at the final step. Then, we can decode the desired entities (中共中央 (CPC Central Committee's), NT), (中国致公党十一大 (the 11th National Congress of the Chinese Zhigong Party), NT) by the semantics of these tags. To the best of our knowledge, we are the first to introduce this iterative denoising process and shift the traditional classification paradigm to the non-autoregressive generation paradigm for sequence labeling.

## 7 Related Work

### 7.1 Sequence Labeling

Sequence Labeling (He et al., 2020) assigns a unique label with special semantics to each unit in a sentence. Named Entity Recognition (Huang et al., 2015b), Part-of-speech tagging (Todi et al., 2018), Chinese Word Segmentation (Shi et al., 2017), Text Chunking (Zhai et al., 2017), Slot Filling (Zhang et al., 2019), Semantic Role Labeling (Daza and Frank, 2018) and many other NLP tasks could be tackled using sequence labeling. Traditional machine learning methods (Baum and Petrie, 1966; Lafferty et al., 2001) and deep-learning-based methods (Huang et al., 2015b; Devlin et al., 2019) are proposed to address it. Recently, knowledge-enhanced pretrained encoders (Diao et al., 2020; Liu et al., 2021; Jiang et al., 2022) boost this field significantly.

### 7.2 Diffusion Model

Diffusion model is derived from the image generation field (Sohl-Dickstein et al., 2015). The strong performance of the diffusion-based generative model (Ho et al. 2020, Ramesh et al. 2021) significantly boosted the development of the image field. Thus, lots of eyes from other fields are attracted by the diffusion model, e.g., speech synthesis (Zhang et al., 2023a), protein design (Trippe et al., 2023), time series forecasting (Li et al., 2022c), natural language generation (Li et al., 2022b). For NLP, the discreteness nature of language symbol poses several challenges. Nonetheless, many techniques are proposed to boost the development of this field (Hoogeboom et al., 2021; Li et al., 2022b; Yuan et al., 2022; Lin et al., 2023; He et al., 2023; Zhou et al., 2023). We provide a more detailed survey about applying the diffusion model to the natural language field in appendix G. Currently, diffusion model dominats the field of generative AI and there are so many works chasing to improve the diffusion model from different per-

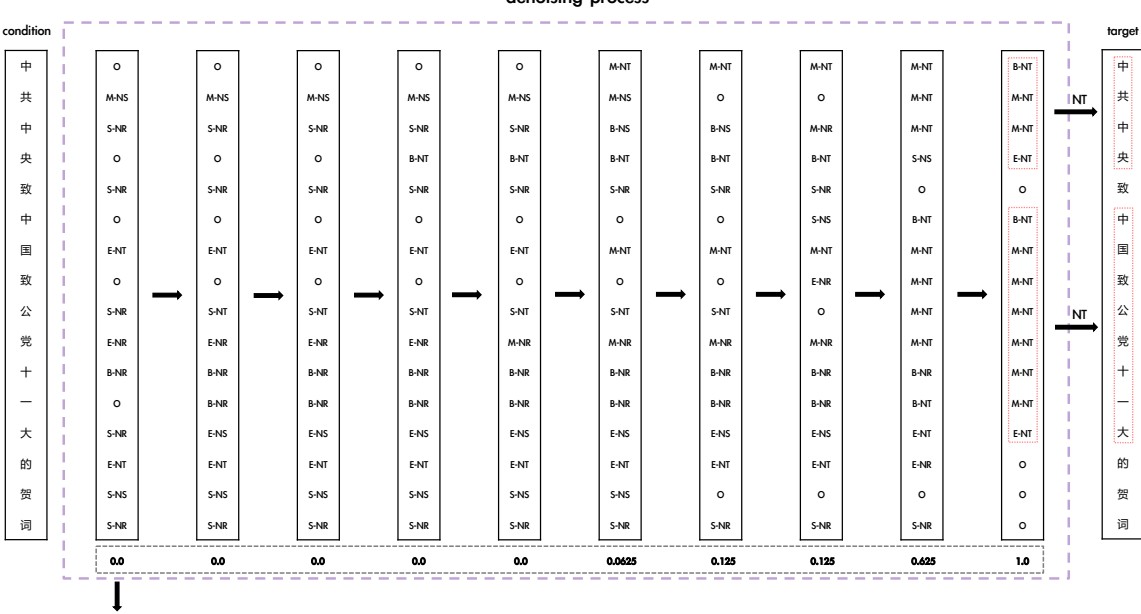

Figure 4: Visulization example for Chinese NER. 中国中央致中国致公党十一大的贺词 means *The CPC Central Committee's congratulatory message to the 11th National Congress of the Chinese Zhigong Party*.

spectives, e.g., likelihood estimation (Kingma et al., 2021), efficient sampling (Song et al., 2021a), condition guided diffusion (Ho and Salimans, 2022).

## 8 Conclusion

In this paper, we cast the Sequence Labeling (SL) task as a conditional generation problem. Furthermore, we introduce a novel framework DiffusionSL to model SL using a non-autoregressive stepwise generation approach. Meanwhile, we propose BT-Converter to handle the discreteness problem and BitDiT to model the progressive noise-removal process. Experimental results indicate that our model performs better on NER, CWS, and POS tagging compared to previous strong baselines, thus revealing the superiority of DiffusionSL. Our work is pioneer research to cast Natural Language Understanding task as a non-autoregressive corresponding generation task. We hope DissusionSL could be used in more tasks that could be formulated as sequence labeling and leave this as the future work, e.g. semantic role labeling, text chunking.

## Limitations

There are several disadvantages that are hard to avoid for DiffusionSL. Firstly, more sampling steps result in a slowerer sampling speed. Though we choose ddim as sampling method to decrease the sampling steps, it is still slower than the discriminative models. Secondly, BitDiT incorporates an extra random-initialized decoder compared to the BERT-style tagger models, which needs more computation memory and is harder to train, we must search the hyperparameters thoroughly to find an optimal result. Thirdly, the initial noise is sampled from a Gaussian distribution, thus bringing randomness in the Tag Diffusion Process.

## Acknowledgement

This work was supported by National Key R&D Program of China (No. 2022ZD0118501) and the National Natural Science Foundation of China (No.U1936207). This work was supported by the Strategic Priority Research Program of Chinese Academy of Sciences (No.XDA27020100), Youth Innovation Promotion Association CAS, and Yunnan Provincial Major Science and Technology Special Plan Projects (No.202202AD080004).

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

## A `int2bit` and `bit2int`

**Algorithm 3** convert tags into bits

```
def tag2bit(x, num_bits):
    mask = 2 ** torch.arange(num_bits - 1, -1,
        -1, dtype=torch.int32)
    x = x.unsqueeze(dim=-1)
    bits = ((x & mask) != 0).float()
    bits = bits * 2 - 1
    return bits
```

**Algorithm 4** convert bits back into tags

```
def bit2tag(x, num_bits):
    x = (x > 0).int()
    mask = 2 ** torch.arange(num_bits - 1, -1,
        -1, dtype=torch.int32)
    tags = (x * mask).sum(dim=-1)
    return tags
```

## B Transformer Block of BitDiT Decoder

We show the architecture details of the Transformer block used in BitDiT decoder in Figure 5. The noisy tag bits $\mathbf{x}_t$ goes through two sub-blocks. The first block consists of an Adaptive Layer Normalization (ALN) layer, a Multi-Head Self-Attention (MHSA) layer and a scaled skip-connection. The second is basically the same but the MSHA layer is substituted by a Feedford Network (FFN) layer. The scale factor before the skip connection is learned based on the conditional embedding $\mathbf{c}_t$. The calculation procudure could be summarized as follows:

$$\mathbf{x}_t = \alpha_1 \times \text{MHSA}(\text{ALN}_1(\mathbf{x}_t), \mathbf{c}_t) + \mathbf{x}_t \quad (19)$$

$$\mathbf{x}_t = \alpha_2 \times \text{FFN}(\text{ALN}_2(\mathbf{x}_t), \mathbf{c}_t) + \mathbf{x}_t \quad (20)$$

$$\alpha_{1,2} = \text{Linear}_{1,2}(\mathbf{c}_t) \quad (21)$$

where $\alpha_1$ and $\alpha_2$ are scale factors. The ALN learns the scale $\gamma$ and shift $\beta$ parameters based on the $\mathbf{c}_t$:

$$\text{ALN}(\mathbf{x}_t, \mathbf{c}_t) = \gamma \times \text{LN}(\mathbf{x}_t) + \beta \quad (22)$$

$$\text{LN}(\mathbf{x}_t) = \frac{\mathbf{x}_t - \mathbb{E}(\mathbf{x}_t)}{\sqrt{\text{Var}(\mathbf{x}_t) + \epsilon}} \quad (23)$$

$$\gamma = \text{Linear}_3(\mathbf{c}_t) \quad (24)$$

$$\beta = \text{Linear}_4(\mathbf{c}_t) \quad (25)$$

where $\epsilon$ is for avoiding division-by-zero error and numerical stability.

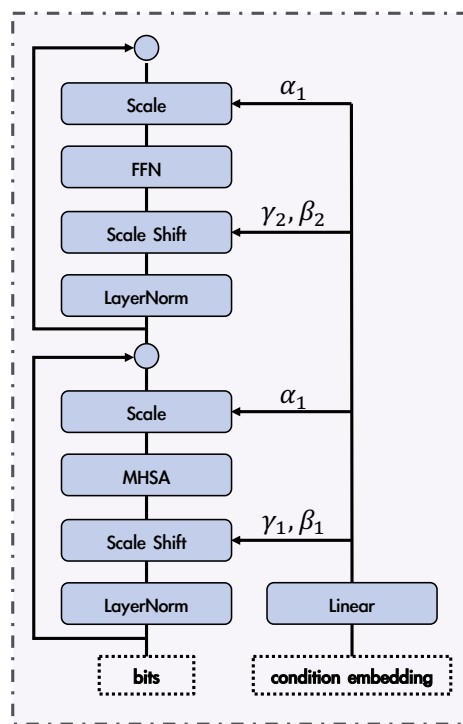

Figure 5: Transformer block details of the **BitDiT** Decoder. MHSA represents Multi-Head Self-Attention and FFN represents Feedforward Network. $\beta$ and $\gamma$ are the learned affine parameters for adaptive layer normalization. $\alpha$ is the scale factor before each skip connection.

## C Statistics of datasets

| Task | Dataset | Train | Dev | Test |
|------|---------|-------|-----|------|
| NER | MSRA | 41728 | 4636 | 4365 |
| | Resume | 3821 | 463 | 477 |
| | Conll03 | 14041 | 3250 | 3453 |
| CWS | MSRA | 78231 | 8693 | 3985 |
| | PKU | 17150 | 1906 | 1944 |
| | CTB6 | 23458 | 2079 | 2796 |
| POS | CTB5 | 18078 | 350 | 348 |

Table 7: The statistics of the datasets. For CWS task, whose dev set is not provided, we randomly select 10% of the original training instances to serve as the dev set.

## D Hyper-parameters settings

For conditional text encoder, we choose `chinese-bert-wwm-ext`[2] for chinese datasets and `bert-large-cased`[3] for English datasetes. We take AdamW(Loshchilov and Hutter, 2019)

[2]https://huggingface.co/hfl/
chinese-bert-wwm-ext
[3]https://huggingface.co/bert-large-cased

as optimizer. We set the peak learning rate as 1e-5, with 1000 warm-up steps and a liner decay scheduler. The timestep for training is set to 1000 and the ddim sampling step is set to 10 for efficiency. The SNR scale is set to 0.1. We use the `B-M-E-S` schema instead of the `B-I-O` schema.

## E Baselines

### E.1 NER

- **BiLSTM-tagger** (Huang et al., 2015b) uses a BiLSTM to encode the sentence and a CRF layer to decode the NER tags.

- **TENER** (Yan et al., 2019) designs a specific attention mechanism and improves the performance of the Transformer for NER.

- **FLAT** (Li et al., 2020) introduces the character-word lattice into the Transformer architecture to enhance the NER performance.

- **NFLAT** (Wu et al., 2022) proposes Inter-Former to apply lexical enhancement and removes the word-character and word-word attention interaction for memory efficiency.

- Athiwaratkun et al. (2020) uses a generative model for joint sequence labeling and sequence classification task. It can tackle many sequence labeling at the same time and performs well in different scenarios.

- Paolini et al. (2021) proposes an augmented natural languages and then cast many sequence labeling tasks as a translation between original language sequence and the augmented ones.

- **UIE** (Lu et al., 2022) utilze the proposed structure extraction language to unify various information extraction tasks and uses unsupervised corpora to pretrain, thus resulting in better information extraction ability.

- **DiffusionNER** (Shen et al., 2023) models the entity as a labeled span (left boundary, right boundary, entity label) and cast the NER as a boundary-denoising process. It generates the entities from the noisy spans using a diffusion model. DiffusionNER sets the boundaries as the generation objective, which can only be used for NER. Conversely, DiffusionSL can be used for all sequence labeling tasks, which enjoys more versatility.

- **gpt-3.5-turbo** (OpenAI, 2023) is one of the most powerful LLMs. We test its one-shot performance to generate the sequence labeling targets, and the corresponding prompts are in Table 6.

### E.2 CWS

- Ma et al. (2018) uses a simple unigram and bigram-based Bi-LSTM to extract the entities.

- Yang et al. (2019) improves the Lattice LSTM by integrating the subword embedding.

- **WMSEG**(Tian et al., 2020) introduces the memory mechanism to incorporate the word-hood information to enhance the segmentation model.

- Cui et al. (2021) enhances the BERT encoder by masking the whole word.

### E.3 POS tagging

For POS tagging, most works focus on adding more features to enhance the word representation ability of the text encoder.

- **ZEN** (Diao et al., 2020) incorporates the n-gram information to enhance the BERT.

- **Glyce** (Meng et al., 2019) incorporates the visual information to enhance the BERT.

- Cui et al. (2021) enhances the BERT encoder by masking the whole word.

## F LLM experiments

We show the prompt we used in the LLM experiments in Table 6. For one-shot experiments, we only add one example following the task definition. For few-shot experiments tested on Resume, we add more demonstration examples following the task definition. For fair comparison in few-shot experiments, examples used in the 15-shot incorporate the ones used in 10-shot, and so on.

## G Detailed Related Work of Diffusion Model with Language

We provide a complete and detailed review of previous work about applying the diffusion model to language for the potential audiences to get familiar with this promising field. Most existing literature focuses on NLG instead of NLU.

- **Multinomial-Diffusion** (Hoogeboom et al., 2021) defines a multinomial diffusion process with categorical data to generate text or a segmentation map.

- **D3PM** (Austin et al., 2021) also defines the diffusion model in discrete space and the newly proposed discrete corruption process improves the performance.

- **DiffusionLM** (Li et al., 2022b) firstly proposes to embed the text into continuous space to circumvent the discreteness problem and round the final denoised vectors back into words.

- **DiffuSEQ** (Gong et al., 2023) firstly employs the diffusion model for seq2sesq text generation setting. And it novelly proposes to concatenate and embed the source and target text and only add noise to the target text embedding while keeping the source text embedding as clean.

- **SSD-LM** (Han et al., 2022) iteratively generates text blocks in a semi-autoregressive manner and conducts the diffusion process on the natural vocabulary space, which balances the advantage of Autoregressive and Non-Autoregressive model and allows convenient guidance incorporation.

- **SED** (Strudel et al., 2022) casts the discrete language symbols into continuous embeddings and incorporates the self-condition trick into the backward denoising process.

- **CDCD** (Dieleman et al., 2022) uses the score-matching framework to solve several language modeling tasks in continuous time and input space.

- **DiffusionBERT** (He et al., 2023) uses the diffusion model to improve the masked language model, which combines the advantage of two denoising models. New proposed noise schedule and time embedding injection methods are applied to it.

- **Difformer** (Gao et al., 2022) proposes an anchor loss function, a layer norm module over the embeddings, and a noise factor that controls the added noise to tackle the problems incorporating the collapse of the denoising objective, the imbalanced norm of embedding

among words, distraction resulted by adding standard noise.

- Lovelace et al. (2022) demonstrates that the latent space of pretrained language model could be used to learn a diffusion model in which the latent representations could be decoded into natural language.

- **SeqDiffSeq** (Yuan et al., 2022) uses a self-condition technique and a newly proposed adaptive noise schedule for sequence-to-sequence diffusion language model based on Transformer.

- **Diff-Glat** (Qian et al., 2022) proposes modality diffusion process and residual glancing sampling to boost the performance of Non-Autoregressive parallel text generation.

- **GENIE** (Lin et al., 2023) pretrains a novel Transformer-based encoder-decoder diffusion language model on a large-scale corpus with a novel pretraining technique named continuous paragraph denoise.

- **DiffusER** (Reid et al., 2023) applies an edit-based generative model based on diffusion denoising process to revise the already existing text, making gradual text refinement probable.

- **RDMs** (Zheng et al., 2023) proposes a novel family discrete diffusion model with a route-and-denoise decoding process.

- **Dinoiser** (Ye et al., 2023a) manipulates the noise in the training and inference stage to avoid the pitfall of discreteness of the embedding space and boost the impact of source sentences for conditional sequence learning.

- **Masked-Diffusion LM** (Chen et al., 2023a) uses a linguistic masking strategy to perturb the clean sentence which enables an easy-first-generation backward process.

- **RenderDiffusion** (Li et al., 2023) transforms the discrete language symbol generation problem into the glyph image generation problem, thus casting the less-studied text-to-text diffusion model as a well-studied text-to-image model.

- **DiffusionSum** (Zhang et al., 2023b) tackles the extractive summary problems by directly

generating the desired summary sentence embedding to match the corresponding natural sentences in the original document.

- Tang et al. (2023) proposes two methods named Distance Penalty and Adaptive Decay Sampling to bridge the gap between the training and inference(namely exposure bias problem in the traditional NLG setting) of the diffusion language model.

- **Diffusion-NAT** (Zhou et al., 2023) unifies the BART and discrete diffusion model and proposes a self-prompting technique for text-to-text generation.

- **AR-Diffusion** (Wu et al., 2023b) combines the Autoregressive language model and the Non-Autoregressive diffusion language model to boost the text generation performance and speed, due to the left-to-right sequential nature of language.

- **DDLM** (Balagansky and Gavrilov, 2023) reproduces the CDCD-based (Dieleman et al., 2022) LM and publicly releases the corresponding training code and checkpoint, then firstly tests the performance of downstream tasks, making it convenient for other researchers to explore this field.

| dataset | prompt |
|---|---|
| MSRA | 请从给定文本中识别人名、地名、组织名并列举出来，每个词最多出现在一个类别中。
文本：我们是受到郑振铎先生、阿英先生著作的启示，从个人条件出发，瞄准现代出版史研究的空白，重点集藏解放区、国民党毁禁出版物。
结果：人名：郑振铎，阿英；地名：无；组织名：国名党
{文本：
结果：} ...
文本：{*text*}
结果： |
| Resume | 请从给定文本中识别国籍、教育水平或学历、地名或籍贯、人名、公司或组织机构名、专业、民族名、职务或身份并列举出来，每个词最多出现在一个类别中。
文本：1966年出生，汉族，中共党员，本科学历，工程师、美国项目管理协会注册会员（PMIMember）、注册项目管理专家（PMP）、项目经理。
结果：国籍：无；教育水平或学历：本科学历；地名或籍贯：无；人名：无；公司或组织机构名：美国项目管理协会；专业：无；民族名：汉族；职务或身份：中共党员，工程师，注册会员，PMIMember，注册项目管理专家，PMP，项目经理；
{文本：
结果：} ...
文本：{*text*}
结果： |
| Conll03 | Please list all Organization, Person, Location, and Miscellaneous Entity in the given text, output using the format as "Entity: Organization: None \| Person: None \| Location: Word1, Word2 \| Miscellaneous: Word3"
Text: The European Commission said on Thursday it disagreed with German advice to consumers to shun British lamb until scientists determine whether mad cow disease can be transmitted to sheep.
Entity: Organization: European Commission \| Person: None \| Location: None \| Miscellaneous: German, British
{ Text:
Entity: } ...
Text: {*text*}
Entity: |

Table 6: The prompt used in LLM experiments. Following Ye et al. (2023b), we use task definition followed by demonstration examples as the prompt. {文本：结果：} ... and {Text: Entity:} ... mean adding more demonstrations. We refer to *text* as the input sentence.