# OpenReview forum: "DiffusionSL: Sequence Labeling via Tag Diffusion Process"
_EMNLP/2023/Conference — EMNLP 2023 Findings_

### Official Review · Reviewer_ckkr · 2023-08-02

**Soundness:** 4

**Excitement:**

4: Strong: This paper deepens the understanding of some phenomenon or lowers the barriers to an existing research direction.

**Paper Topic And Main Contributions:**

The paper tackles sequence labeling tasks by using a diffusion approach. The authors design a diffusion transformer that models the diffusion process and a Bit-Tag converter to handles the discrete tags in continuous space. The approach is evaluated on three tasks with good results.

**Questions For The Authors:**

A) How does the initial sampling of random tags impact performance? Have you tried to do some experiments to assess that?

B) Does the number of possible tags affect the effectiveness of the model? In all the selected tasks the tag vocabulary is small, I am curious if you have tried to look into this factor.

C) Are the results reported the average of more runs?

D) What are the computational performance of the method? Something is mentioned in the limitations section, a brief discussion or table in the main text would be interesting.

**Reasons To Accept:**

- The paper introduces a new framework that tackles sequence labeling tasks using a transformer with the diffusion process instead of the standard autoregressive approach.

- Strong experimental setup with good results and comparisons

- Paper is clear and well written

**Reasons To Reject:**

- Experimental setup is described a bit too briefly and some small details might be missing, however it can be easy to fix.

**Reproducibility:**

3: Could reproduce the results with some difficulty. The settings of parameters are underspecified or subjectively determined; the training/evaluation data are not widely available.

**Reviewer Confidence:**

4: Quite sure. I tried to check the important points carefully. It's unlikely, though conceivable, that I missed something that should affect my ratings.

**Typos Grammar Style And Presentation Improvements:**

- Experimental setup description should be expanded a bit, describing for example what are the performance metrics which are not mentioned

- I would suggest the authors consider including the results of a simple vanilla baseline like BERT.

- It would be better if the algorithms shown would be in pseudocode

- There are some typos here and there (e.g DiffusionSL is misspelled at the end of the conclusions)

---

> ### Author Rebuttal · Authors · 2023-08-27
>
> Thanks for taking the time to evaluate our paper and giving your careful and insightful comments and suggestions. Moreover, thanks for your affirmation to our paper. We have carefully considered the reviews and would like to address the concerns and comments raised.
>
> * **Response to `Reasons To Reject`:** Thanks for bringing up this problem. We provide more details in the appendix C, D, and E. We will add more experimental setup details in the revised version to make our paper clearer.
>
> * **Response to Qeustion A:** We use torch.randn_like() as the sampling method. And we conduct experiments using different random seeds to report the averge results. We find the performance is robust agsinst the randomness of sampling. We will add more detailed descriptions in the revised version.
> * **Response to Question B:** Yes, we have considered this problem. And we conduct experiments to test the effectiveness against the nubmer of tags. In Table 4, we try to test two methods converting the discrete tags into continuous representation. The first one is using bit representation (BTConverter in the paper), and the second one is learning a embedding for each tag. NER-Resume has 13 tags (dimension is 4, 2^4=16) and CTB5 has 103 tags (dimension is 7, 2^7=128). For bit-representation, it shows strong performance for small and big vocabulary. For embedding-representation of same dimension with bit-representation, it shows that the performance of small vocabulary (13 tags, dimension is 4) does not converge. Increasing the dimension of embedding-representation to 768 also dose not increse the performance dractically. So we acclaim the superiority of our bit-representation method.
>
> * **Response to Question C:** Yes, we report the averge runs. The performance is robust.
> * **Response to Question D:** Thanks for bringing up this idea. We would like to add some discussion in the revised version. For prior experiments, the training and inference speed of DiffusionSL is faster than DiffusionNER at least.
> * **Response to Typos Grammar Style And Presentation Improvements:** Thanks for evaluating our paper carefully. We would add more experiments setup details, add simple BERT baseline, try to describe the pseudocode in tradiitonal style instead of pytorch-style, and fix the typos in the paper.

---

### Official Review · Reviewer_kB8h · 2023-08-06

**Soundness:** 3

**Excitement:**

3: Ambivalent: It has merits (e.g., it reports state-of-the-art results, the idea is nice), but there are key weaknesses (e.g., it describes incremental work), and it can significantly benefit from another round of revision. However, I won't object to accepting it if my co-reviewers champion it.

**Paper Topic And Main Contributions:**

This paper discusses a framework called DiffusionSL that uses Diffusion models for sequence labeling tasks. Moreover to incorporate the discrete nature of NL tokens, they introdcued Bit-Tag converter (for quantization) and Bit Diffusion Transformer (for noise removal).

**Questions For The Authors:**

1. Is it possible to do any nested NER tasks in the proposed approach? Is it possible to see results on ACE04 and ACE05?
2. I think it will be interesting to see the performance in any few-shot setting with diffusion approach.

**Reasons To Accept:**

1. The proposed method shows some performance gains over other methods in this domain.
2. Besides diffusion modeling like previous works, the authors also tries to address the quantization of tokens.

**Reasons To Reject:**

1. This work iteratively improves over DiffusionNER both using similar techniques. However, DiffusionNER also focuses on span level information which DiffusionSL does not directly address. I am not sure whether that is a strength or weakness for DiffusionNER vs. DiffusionSL.

2. I am also not sure how span level information can be handled in current framework of DiffusionSL.

3. Related to 2, I think nested NER tasks are missing in the evaluation.

4. (After rebuttal) Seems like DiffusionSL is less generalizable than DiffusionNER being not applicable to nested NER which objectively is a more difficult task.

**Reproducibility:**

4: Could mostly reproduce the results, but there may be some variation because of sample variance or minor variations in their interpretation of the protocol or method.

**Reviewer Confidence:**

3: Pretty sure, but there's a chance I missed something. Although I have a good feel for this area in general, I did not carefully check the paper's details, e.g., the math, experimental design, or novelty.

---

> ### Author Rebuttal · Authors · 2023-08-27
>
> Thanks for taking the time to evaluate our paper and giving your careful and insightful comments and suggestions. We have carefully considered the reviews and would like to address the concerns and comments raised.
>
> * **Response to `Reasons To Reject`1:**
>   * DiffusionSL is not an incremental work over DiffusionNER. There are many difference: DiffusionSL focuses on sequence labeling instead of only NER, which is modeled as a conditional discrete tag data generation problem in our framework. We conduct experiments on POS tagging and Chinese Word Segmentation datasets to test the effectiveness on different sequence labeling tasks other than NER. DiffuisonSL could be extended to other sequence labeling tasks easily, such as slot filling. We leave this as future work. DiffusionSL utilizes the bit-representation as an explicit solution to discreteness problem, which could be beneficial and insightful for more NLP tasks (Most NLP tasks are inherently discrete tasks). DiffusionSL uses an specific Transformer instead of the vanilla Transformer, which shows faster convergence (Fig 3).
>   * Yes, DiffusionSL does not directly focuses on span level informaiton. However, we do not think it is the weakness of DiffusionSL. We think the generation format of DiffusionNER (e.g., span boundary) is unnatural, and it separates the span localization and the entity label classification to some extent. Thus the diffusion model (novel and effective part of DiffusionNER) can not directly model the entity label information (the generation target x_0 is just boundary,  and DiffusionNER uses another MLP Classifier to incorporate the span label information) well. What's more, this generation format necessitates extra heuristic post-propocessing operations. Though DiffusionNER considers the span-level information, we think this modeling style also incorporates the above shortcomings. And the experiment results demonstrate the superiority of DiffusionSL over DiffusionNER in flat NER datasets.
> * **Response to `Reasons To Reject`2:** For span level information, we do not explicitly model it in the DiffusionSL. But it could handle span-level tasks well. Sequence labeling is inherently such method to convert the span-level information and token-level information back and forth equvalently. The span-level and token-level information conversion is a bijection.
> * **Response to `Reasons To Reject`3:** DiffusionSL could not be directly extended to nested NER and discontinuous NER (DiffusionNER is also not suitable for discontinuous NER) now. The intuitive way to extend the DiffusionSL to nested NER is to extend the BIO format to BILOU hypergraph-based labeling schema. With more novelty, we could extend the 1D labeling schema (BIO format) to 2D labeling schema (like BiaffineNER[1] or W2NER[2]) for nested NER or even discontinuous NER. We think your suggestion is well, and we leave this idea as a future work to explore.
> * **Response to Question1:** As we mentioned in response to reject3, the current version of DiffusionSl could not be extended to nested NER. We leave this as future work.
> * **Response to Question2:** Most of the NER studies focuses on fully-supervised setting, including the DiffusionNER. For fair comparison with previous methods, we also adopt the fully-supervised setting. However, it is a good research direction to explore the few-shot performance of diffusion model for NLU tasks. We randomly sample 5/10/20 examples from Resume (NER dataset) for training and provide the corresponding experimental results. Thanks for your insightful comments.
>
> | Number of Samples |   P   | R     | F1    |
> | ----------------- | :---: | ----- | ----- |
> | 5                 | 29.35 | 10.96 | 15.96 |
> | 10                | 62.64 | 50.15 | 55.71 |
> | 20                | 80.61 | 71.83 | 75.97 |
>
> Thanks for taking time to read these. We sincerely hope you could consider raising the scores. We are also expecting more discussions about DiffusionSL.
>
> [1]Yu, Juntao, Bernd Bohnet, and Massimo Poesio. "Named entity recognition as dependency parsing. "ACL2020
>
> [2]Li, Jingye, et al. "Unified named entity recognition as word-word relation classification."AAAI2022

---

### Official Review · Reviewer_zKAW · 2023-08-07

**Soundness:** 3

**Excitement:**

3: Ambivalent: It has merits (e.g., it reports state-of-the-art results, the idea is nice), but there are key weaknesses (e.g., it describes incremental work), and it can significantly benefit from another round of revision. However, I won't object to accepting it if my co-reviewers champion it.

**Missing References:**

[1] Chen T, Zhang R, Hinton G. Analog Bits: Generating Discrete Data using Diffusion Models with Self-Conditioning[C]//The Eleventh International Conference on Learning Representations. 2022.

**Paper Topic And Main Contributions:**

Paper Topic: Sequence Labeling

Main Contribution: The paper formulates the Sequence Labeling task as a conditional generation problem, where diffusion modeling is adopted. To overcome the discreteness problem, the paper introduces bit-format inputs with Bit-Tag Converter and a Bit Diffusion Transformer. Experimental results demonstrate the validity of the proposed approach.

**Questions For The Authors:**

- One trivial question: What's the advantage of generative modeling over discriminative modeling on the sequence labeling problem?

**Reasons To Accept:**

- The paper is well-written and easy to follow. The problem is well-formulated.
- The idea is interesting and the design is reasonable to apply diffusion models to the sequence labeling problem.
- Experiments are sufficient, and ablation studies are conducted to check the effectiveness of the proposed method.

**Reasons To Reject:**

- **The proposed approach (i.e., Bit Diffusion Transformer) is similar to [1]**, but the authors did not include any discussion or citation. Since [1] share a very similar design to the proposed one. Therefore, this would result in the novelty problem directly. Furthermore, **the authors did not cite it**.

[1] Chen T, Zhang R, Hinton G. Analog Bits: Generating Discrete Data using Diffusion Models with Self-Conditioning[C]//The Eleventh International Conference on Learning Representations. 2022.

**Reproducibility:**

4: Could mostly reproduce the results, but there may be some variation because of sample variance or minor variations in their interpretation of the protocol or method.

**Reviewer Confidence:**

4: Quite sure. I tried to check the important points carefully. It's unlikely, though conceivable, that I missed something that should affect my ratings.

---

> ### Author Rebuttal · Authors · 2023-08-27
>
> Thanks for taking the time to evaluate our paper and giving your careful and insightful comments and suggestions. We have carefully considered the points raised in the reviews and would like to address the concerns and comments raised.
>
> * **Response to `Reasons To Reject`:**
>   * We regret the oversight in not citing [1]. In line 110~113, we discuss and cite two existing representative diffusion models that convert the discrete data into continuous data for tackling the discreteness problem. One is learning a embedding for each discrete class and the other is the bit-representation ([1] is the first to propose this line of work and [2] is another application work) method. Based on the bit-representation method, **our novelty and contribution are modeling the sequence labeling problem as a conditional discrete tag generation problem and using a non-autoregressive step-wise model with self-correction ability for discrete data generation.** What's more,  introducing the bit-representation for discrete label to connect the sequence label/tag data and Gaussian diffusion model (in continuous data space), and using a specific devised bit diffusion Transformer (as you mentioned in review and we introduce it in section 3.3) instead of the vanilla Transformer or UNet in [1] as the generation backbone to utilize and harness the strong generation ability of diffusion model are model-component-level contribution.
>   * It is noteworthy that we cite [2] in line 112, which is a work using [1] directly to solve the panoptic segmentation problem and even enjoying the same first-author with [1]. So please believe that we are unintentional to ommit the citation. We originally cite both [1] and [2] in the earliest draft and the frequent draft-editing by different authors might lead to the missing problem. And we will add the citation back in the revised version.
>   * We provide detailed and strong ablation studies to support the advantage of  component design (BTConverter and BitDiT) over others, e.g., lightweight, flexible for BTConverter and faster convergence and better performance for BitDiT. We will provide more motivation discussion in the introduction section to clarify the necessity that we design these components in the revised version. From our perspective, sequence labeling is almost one of the most popular paradigm to tackle various NLP tasks, so introducing a completely different modeling style (non-autoregressive generation with self-correction) for sequence labeling brings more research opportunities for the future work, and thus it does not lack novelty.
>   * Based on the above points, we sincerely hope you could consider raising the scores. We expect more discussion about the details in DiffusionSL.
>
> * **Response to Question:** In general, we can train generative model to output in any specific formats by teacher forcing. Generative methods enjoy the properties of flexibility and generality (as mentioned in [3])  to tackle various tasks against discriminative ones. What's more, many generative baselines demonstrate strong performance. We believe the potential of generative models for sequence labeling has not been fully tapped. So we introduce another line of generative methods into sequence labeling. The non-autoregressive manner enables the diffusion model to self-correct the generation results in the denoising path, which is a advantage the discriminative methods do not have. Moreover, the DiffusionSL could model the interactions between the tags. Conversely, the discriminative methods just independently predict the tag for each token and do not explicitly model the relations between the tags.
>
> [1] Chen T, Zhang R, Hinton G. Analog Bits: Generating Discrete Data using Diffusion Models with Self-Conditioning
>
> [2] Chen, T., Li, L., Saxena, S., Hinton, G., & Fleet, D. J. (2022). A generalist framework for panoptic segmentation of images and videos.
>
> [3] Raman, Karthik, et al. Transforming sequence tagging into a seq2seq task.

---

### Meta-Review · Area_Chair_2yzD · 2023-09-19

**Recommendation:** 3

**Metareview:**

The paper frames the Sequence Labeling task as a conditional generation challenge, utilizing diffusion modeling. In addressing the issue of discreteness, the paper presents a solution involving bit-format inputs using the Bit-Tag Converter and a Bit Diffusion Transformer. It omits a pertinent citation, namely "Analog Bits: Generating Discrete Data using Diffusion Models with Self-Conditioning." A more comprehensive discourse and experimental evaluation comparing the proposed approach with prior systems, including the absent reference and DiffusionNER, is warranted

---

### Decision · Program_Chairs · 2023-10-07

**Decision:**

Accept-Findings

**Comment:**

The paper frames the Sequence Labeling task as a conditional generation challenge, utilizing diffusion modeling. In addressing the issue of discreteness, the paper presents a solution involving bit-format inputs using the Bit-Tag Converter and a Bit Diffusion Transformer. It omits a pertinent citation, namely "Analog Bits: Generating Discrete Data using Diffusion Models with Self-Conditioning." A more comprehensive discourse and experimental evaluation comparing the proposed approach with prior systems, including the absent reference and DiffusionNER, is warranted